# Research on Attitude Detection and Flight Experiment of Coaxial Twin-Rotor UAV

**DOI:** 10.3390/s22249572

**Published:** 2022-12-07

**Authors:** Deyi You, Yongping Hao, Jiulong Xu, Liyuan Yang

**Affiliations:** 1School of Equipment Engineering, Shenyang Ligong University, No.6, Nanping Central Road, Hunnan New District, Shenyang 110159, China; 2Dezhou Vocational and Technical College, Dezhou 253034, China

**Keywords:** rotor, aircraft, control, system, software, PID, attitude, filtering, unmanned aerial vehicle (UAV)

## Abstract

Aiming at the problem that the single sensor of the coaxial UAV cannot accurately measure attitude information, a pose estimation algorithm based on unscented Kalman filter information fusion is proposed. The kinematics and dynamics characteristics of coaxial folding twin-rotor UAV are studied, and a mathematical model is established. The common attitude estimation methods are analyzed, and the extended Kalman filter algorithm and unscented Kalman filter algorithm are established. In order to complete the test of the prototype of a small coaxial twin-rotor UAV, a test platform for the dynamic performance and attitude angle of the semi-physical flight of the UAV was established. The platform can analyze the mechanical vibration, attitude angle and noise of the aircraft. It can also test and analyze the characteristics of the mechanical vibration and noise produced by the UAV at different rotor speeds. Furthermore, the static and time-varying trends of the pitch angle and yaw angle of the Kalman filter attitude estimation algorithm is further analyzed through static and dynamic experiments. The analysis results show that the attitude estimation of the UKF is better than that of the EKF when the throttle is between 0.2σ and 0.9σ. The error of the algorithm is less than 0.6°. The experiment and analysis provide a reference for the optimization of the control parameters and flight control strategy of the coaxial folding dual-rotor aircraft.

## 1. Introduction

The measurement accuracy of the attitude information of a coaxial UAV directly affects the navigation accuracy of the speed and position. The gyroscopes, accelerometers and magnetometers are often used as attitude measurement devices. Gyroscopes have cumulative errors, while accelerometers and magnetometers are subject to greater external interference. Effective data fusion is required to ensure a more accurate attitude horn. The flight principle of coaxial twin-rotors is similar to that of a common helicopter. Two coaxial blades with the same diameter, but rotating in opposite directions, are used to control the pitching and rolling attitude of the aircraft by changing the pitch of the tilting disk. However, unlike helicopters, coaxial twin-rotors cancel the tail rotor used to balance the yawing moment on the helicopter and realize the yawing motion of the aircraft through the differential control of the twin-rotors. With the continuous development of power, automatic control and micro-system technology, multi-axis rotor aircraft represented by four-axis rotors have achieved great success. Rotors have been widely used in consumer drones, plant protection, aerial photography and other industries [1,2,3].

The UAV has the advantages of small size and high aerodynamic efficiency. However, the inner and outer shafts of the coaxial upper and lower rotors are commutated by the bevel gear of the main reducer. It has a complex structure and control mechanism, as well as complicated vibration characteristics and nonlinearity. The helicopter rotor rotation test-bed designed by Wu Zhigang is used to measure the noise when the rotor rotates. The anechoic chamber creates a completely anechoic test environment without affecting the rotor flow field, which can meet the test requirements of the model rotor noise radiation characteristics of the helicopter main rotor and tail rotor, and provides an ideal test facility for studying helicopter rotor noise. Zhang Donglin analyzed the inherent characteristics and sensitivity of coaxial counter-rotating closed differential gear trains. The correctness of the theoretical analysis is verified by an example calculation, which provides a basis for the inherent characteristic analysis and optimal design of coaxial counter-rotating closed differential gear trains. An experimental study by Prasad Rao Jubilee analyzes the influence of the different motor duty ratio and propeller pitch value on motor propeller systems with two to four coaxial rotors. The results show that, in the dual-rotor coaxial system, in order to reduce the adverse effects of the front rotor backwashing and run at the best performance, only the initial work of the rear motor should reach 75% duty cycle before the front motor reaches 75% duty cycle. Additional thrust requirements should be generated from the rear rotor and then from the front rotor until its maximum duty cycle [4].

The yawing of UAVs is realized by the differential speed of the upper and lower propellers in producing different torques. The up-and-down power of the UAV is transmitted by gear reduction transmission. Therefore, the mechanical noise and vibration of an unmanned aerial vehicle (UAV) in flight are inevitable. This noise and vibration will have great influence on MEMS accelerometers and sensors. The UAV’s attitude and control are directly affected during actual flight. Consequently, an effective data filtering method can better control the attitude of the UAV, and then the position information of the UAV can be determined in order to control the UAV to fly autonomously. Adam Bondyra introduced the development of a comprehensive data-driven FDI method for the rotor failure of multi-rotor UAV. The algorithm aims at early detection of damaged rotor faults, which lead to the decline of flight stability, decrease the safety of UAV operation and increase the power consumption of the power train. The paper analyzes the kinematics and dynamics characteristics for the coaxial attitude estimation method and establishes a hardware-in-the-loop simulation attitude angle test platform to analyze the mechanical vibration and attitude angle performance of UAVs. For the experimental analysis of the mechanical vibration characteristics of different rotor speeds, the EKF and UKF filtering algorithm models are established, and the trend of attitude changes over time is verified in static and dynamic conditions, respectively. The UKF attitude estimation method is better than the EKF attitude estimation method, and the test flight experiment verifies that the proposed algorithm has better attitude estimation results for the coaxial UAV.

## 2. Modeling of Coaxial UAV

The mathematical model of coaxial UAV is the basis of flight control and navigation method research. In order to enhance the trueness of the model in the control system simulation, a 6-DOF dynamic equation is established. In order to realize the flight balance and movement of the co-axial twin-rotor aircraft, it is necessary to have a mathematical model of the system first, and then analyze the characteristics and control of the system. Therefore, the establishment of the mathematical model has a particularly important influence on the control of the whole system [5,6,7,8]. According to the Newton Euler formula, the state equation describing the motion characteristics of the co-axial twin-propeller aircraft is derived, and the mathematical model of the aircraft is established. As can be seen from the following Figure 1, the common axis dual-rotor UAV is mainly composed of upper and lower rotors, fuselage, control panel, control mechanism and controller battery core cabin. The lift is generated by the up-and-down counter-rotor rotation of the UAV. The structural model diagram is shown in figure. The lateral movement generates the lateral force component through the tilt of the control panel, and the control quantity includes four input quantities: throttle input, roll, pitch and yaw [9,10]. The structural model diagram of coaxial twin-rotor UAVs includes the kinematic model and dynamic model, as shown in figure.

The whole system of the coaxial twin-rotor UAV is complex, and the coupling degree between modules is high. In order to better analyze the dynamics and kinematics mathematical model of coaxial twin-rotor UAVs, the coaxial twin-rotor UAV is divided into several subsystems. Formula 1 is the nonlinear model.
(1)$$\dot{X}=f\left(x\;u\;w\right)$$

### 2.1. Dynamics Model of a Coaxial Twin-Rotor UAV

Before deriving the dynamic model of the co-axial twin-rotor UAV, it is assumed that the UAV is a rigid body, its center of mass is on the central axis of the UAV, and the whole UAV is completely symmetric. The six degrees of freedom kinematic equation and the Newton-Euler dynamic equation follow. The dynamic equation of UAV can be expressed as Equation (2).
(2)$$\left\{ \begin{array}{l} \sum \vec{F}=m\frac{d\vec{V}}{dt}\\ \sum \vec{M}=\frac{d\vec{L}}{dt} \end{array}\right.$$

In the abovementioned formula, $\vec{F}$ is the resultant force of the external force of the coaxial dual-rotor UAV. $\vec{V\;}$ is the linear velocity at the center of mass. *m* is the total mass of the rotor UAV. *M* is the total moment of external force. *L* is the total momentum moment. The above-mentioned dynamic equations are obtained in the static coordinate system, but the UAV is always moving when it performs its mission; thus, the dynamic equations in the dynamic coordinate system should be considered when studying the dynamics of UAV. According to Euler’s equation of motion, the absolute derivative in dynamic coordinate systems can be obtained as follows.
(3)$$\left\{ \begin{array}{l} \sum \vec{F}={\left(m\frac{d\vec{V}}{dt}\right)}_{rot}+\vec{\Omega }\times \vec{P}\Leftrightarrow \sum \vec{F}=m\left(\vec{V}+\vec{\Omega }\times \vec{V}\right)\\ \sum \vec{M}={\left(\frac{d\vec{L}}{dt}\right)}_{rot}+\vec{\Omega }\times \vec{L}\Leftrightarrow \sum \vec{M}=\vec{I}\cdot \vec{\Omega }+\vec{\Omega }\times \left(\vec{I}\times \vec{\Omega }\right) \end{array}\right.$$

Combining the dynamic equation with the motion of coaxial twin-rotor UAVs, the matrix expression of its dynamics can be obtained as follows:(4)$$\left[\begin{array}{cc} mI_{3\times 3} & 0_{3\times 3}\\ 0_{3\times 3} & I \end{array}\right]\left[\begin{array}{c} {\dot{V}}_{b}\\ {\dot{\omega }}_{b} \end{array}\right]=\left[\begin{array}{c} F_{b}\\ M_{b} \end{array}\right]-\left[\begin{array}{c} \omega_{b}\times \left(mV_{b}\right)\\ \omega_{b}\times \left(I\omega_{b}\right) \end{array}\right]$$
where $V_{b}$ represents the speed of the coaxial dual-rotor UAV in the body coordinate system relative to the geographic coordinate system; $\omega_{b}$ represents the coaxial dual-rotor UAV in the body coordinate system relative to the geographic coordinate system angular velocity; $I_{3\times 3}$ represents the unit matrix of the moment of inertia; I represents the moment of inertia of the coaxial dual-rotor drone; $F_{b}\;$represents the external force on the coaxial dual-rotor drone in the body coordinate system combined; $M_{b}$ represents the external moment that the coaxial dual-rotor UAV in the body coordinate system is combined. In order to facilitate the calculation and understanding in the future, the linear motion of the UAV is represented by the geographic coordinate system, and the attitude change of the UAV is represented by the body coordinate system. Thus, the formula above (4) can be changed as follows.
(5)$$\left[\begin{array}{cc} mI_{3\times 3} & 0_{3\times 3}\\ 0_{3\times 3} & I \end{array}\right]\left[\begin{array}{c} {\dot{V}}_{n}\\ {\dot{\omega }}_{b} \end{array}\right]=\left[\begin{array}{c} F_{n}\\ M_{b} \end{array}\right]-\left[\begin{array}{c} 0_{3\times 3}\\ \omega_{b}\times \left(I\omega_{b}\right) \end{array}\right]$$

During the flight of the coaxial twin-rotor UAV, the resultant force is mainly composed of three forces, which are the lift generated by the rotation of the blades, the gravity of the UAV itself and the air resistance during the flight of the UAV. In the airframe coordinate system, the force formula of UAV is as follows in Formula (6).
(6)$$\sum F_{b}=F_{T}-F_{G}+C_{n}^{b}F_{D}$$
where $C_{n}^{b}$ is the rotation matrix. The lift generated by the upper and lower blades of the UAV is $F_{1}\;$and$\;F_{2}$ respectively, and the direction of the generated lift is perpendicular to the blade tip. When the UAV is flying in an environment with no wind or low wind speed, the influence of air on the UAV’s flight attitude can be neglected. The lift force of the UAV in hovering state is shown in Formula (7).
(7)$$F_{b,T}=\left[\begin{array}{c} 0\\ 0\\ F_{1}+F_{2z} \end{array}\right]$$ where$\;F_{1}$, $F_{2}$ are the lift generated by the rotation of the upper and lower blades. $F_{2z}$ represents the projection of the lift generated by the lower blade on the *z*-axis. The lift generated by the rotation of the blades can be calculated using the following formula.
(8)$$F_{T}=b\omega^{2}$$
where b represents the Blade lift coefficient. ω represents the rotational angular velocity of the blade.

### 2.2. Kinematic Model of Coaxial Twin-Rotor UAV

From the above definition, we can determine that the speed of the coaxial dual-rotor UAV in the body coordinate system is $V_{b}={\left[u\;v\;w\right]}^{T}$. Convert it to the speed in the Earth coordinate system through the rotation matrix and express it as follows.
(9)$$\left\{ \begin{array}{l} \dot{x}=u\cos\theta \cos\psi +v\left(\sin\theta \sin\phi \cos\psi -\cos\phi \sin\psi \right)+w\left(\sin\theta \cos\phi \cos\psi +\sin\phi \sin\psi \right)\\ \dot{y}=u\cos\theta \sin\psi +v\left(\sin\theta \sin\phi \sin\psi +\cos\phi \cos\psi \right)+w\left(\sin\theta \cos\phi \sin\psi -\sin\phi \cos\psi \right)\\ \dot{z}=-u\sin\theta +v\sin\phi \cos\theta +w\cos\phi \cos\theta \end{array}\right.$$

The relationship of the line motion of the drone was discussed above. Next, using the previous rotation formula to convert the angular velocity of the coaxial dual-rotor UAV in the body coordinates into the angular velocity of the geographic coordinate system is as follows.
(10)$$\omega_{b}=\left[\begin{array}{c} p\\ q\\ r \end{array}\right]=R\left(x,\phi \right)R\left(y,\theta \right)\left(\begin{array}{c} 0\\ 0\\ \dot{\psi } \end{array}\right)+R\left(x,\phi \right)\left[\begin{array}{c} 0\\ \dot{\theta }\\ 0 \end{array}\right]+\left[\begin{array}{c} \dot{\phi }\\ 0\\ 0 \end{array}\right]=\left[\begin{array}{ccc} 1 & 0 & -\sin\theta \\ 0 & \cos\phi & \sin\phi \cos\theta \\ 0 & -\sin\phi & \cos\phi \cos\theta \end{array}\right]\left[\begin{array}{c} \dot{\phi }\\ \dot{\theta }\\ \dot{\psi } \end{array}\right]$$

Equation (10) can also be transformed into the following.
(11)$$\left\{ \begin{array}{l} \dot{\phi }=p+\left(r\cos\phi +q\sin\phi \right)\tan\theta \\ \dot{\theta }=q\cos\phi -r\sin\phi \\ \dot{\psi }=\frac{1}{\cos\theta }\left(r\cos\phi +q\sin\phi \right) \end{array}\right.$$

In the kinematics model of the UAV, when the coaxial twin-rotor UAV is flying at a slow speed or hovering in the air, the air resistance of the airframe is very small and can be ignored. Moreover, because the blade selected by the UAV is made of carbon fiber, and its weight is relatively light, the gyro effect produced by the blade can be neglected, which will not affect the flight of the UAV.

The designed coaxial twin-rotor UAV is a nonlinear system with four inputs and six outputs. Its four inputs are the rotational speed of the upper and lower blades and two steering gears. In order to establish a controller for its system model and its characteristics, the system inputs are defined as Formula (12):(12)$$\left\{ \begin{array}{l} U_{1}=F_{1}+F_{2z}\\ U_{2}=M_{Tx}\\ U_{3}=M_{Ty}\\ U_{4}=M_{up}-M_{lw} \end{array}\right.$$
where *U*_1_ is the height control quantity; *U*_2_ is the pitch control quantity; *U*_3_ is the roll control quantity; *U*_4_ is yaw control.

Formula (12) is merged with Formula (9) to obtain the following mathematical model of coaxial twin-rotor UAVs:(13)$$\left\{ \begin{array}{l} \ddot{x}=\left(\sin\theta \cos\phi \cos\psi +\sin\phi \sin\psi \right)\frac{U_{1}}{m}\\ \ddot{y}=\left(\sin\theta \cos\phi \sin\psi -\sin\phi \sin\psi \right)\frac{U_{1}}{m}\\ \ddot{z}=\frac{U_{1}}{m}\cos\phi \cos\theta -g \end{array}\right.$$

Combined with the above requirements, the general nonlinear system model of coaxial UAVs is as follows.
(14)$$\left\{ \begin{array}{l} \ddot{\phi }=\frac{I_{y}-I_{z}}{I_{x}}\dot{\theta }\dot{\psi }+\frac{U_{2}}{I_{x}}\\ \ddot{\theta }=\frac{I_{z}-I_{x}}{I_{y}}\dot{\phi }\dot{\psi }+\frac{U_{3}}{I_{y}}\\ \ddot{\psi }=\frac{I_{x}-I_{y}}{I_{z}}\dot{\phi }\dot{\theta }+\frac{U_{4}}{I_{z}} \end{array}\right.$$

### 2.3. Attitude Estimation Methods

Common attitude estimation methods include the Mahony complementary filtering algorithm, extended Kalman filtering algorithm, etc. Aiming at the structural characteristics of coaxial twin-rotor UAVs and considering the sensor noise, the extended Kalman filtering algorithm and unscented Kalman filtering algorithm models are established [11]. The established nonlinear equation is shown in Equation (15).
(15)$$\{\begin{array}{l} x_{k}=f(x_{k-1},u_{k})+\omega_{k}\\ z_{k}=h(x_{k-1})+v_{k} \end{array}$$
where $\omega_{k}\sim \left(0,Q_{k}\right)$, $v_{k}\sim \left(0,R_{k}\right)$. Considering the discretization equation of the nonlinear system, Equation (16) can be expressed as follows.
(16)$$\{\begin{array}{l} x(k)=f(x(k-1),k-1)+w(k-1)\\ z(k)=h(x(k),k)+v(k) \end{array}$$
where x(k) is the state matrix of the time system at time *k*; z(k) is the observation matrix at time *k*; f(x(k),k) and h(x(k),k) are measurement state functions, which represent the predicted state of the nonlinear discrete system. Here, it is assumed that the state quantity of the system is $x\left(k\right)=\left(q_{0}\left(k\right)\;q_{1}\left(k\right)\;q_{2}\left(k\right)\;q_{3}\left(k\right)\;b_{wx}\left(k\right)\;b_{wy}\left(k\right)\;b_{wz}\left(k\right)\right)$, where x(k) includes the attitude quaternion and random drift quantity of three rotating axis gyroscopes.
(17)$$\left[\begin{array}{c} {\dot{q}}_{0}\\ {\dot{q}}_{1}\\ {\dot{q}}_{2}\\ {\dot{q}}_{3} \end{array}\right]=\frac{1}{2}\left[\begin{array}{cccc} 0 & -\omega_{nb}^{bx} & -\omega_{nb}^{by} & -\omega_{nb}^{bz}\\ \omega_{nb}^{bx} & 0 & \omega_{nb}^{bz} & -\omega_{nb}^{by}\\ \omega_{nb}^{by} & -\omega_{nb}^{bz} & 0 & \omega_{nb}^{bx}\\ \omega_{nb}^{bz} & \omega_{nb}^{by} & -\omega_{nb}^{bx} & 0 \end{array}\right]\cdot \left[\begin{array}{c} q_{0}\\ q_{1}\\ q_{2}\\ q_{3} \end{array}\right]$$

In order to better analyze the data processed by the computer, the above formula is discretized by the Runge-Kutta method of order 4, and the simplified equation after discretization is as follows.
(18)q(k)=F(k−1)×q(k−1) where F(k−1) is the following expressions:(19)$$F(k-1)=I+\frac{T}{2}M(\omega )+\frac{T^{2}}{8}M^{2}(\omega )+\frac{T^{3}}{48}M^{3}(\omega )+\frac{T^{4}}{384}M^{4}(\omega )$$

Because of MEMS gyroscope drift wide randomness, compensation is not easy. Therefore, we will decide the gyroscope drift, as state variables, and quaternions together to form the final state variables.
(20)$$x={\left[q_{0},q_{1},q_{2},q_{3},\beta_{x},\beta_{y},\beta_{z}\right]}^{T}$$

The gyroscope random drift satisfies the following measurement value updated equation, as shown in the following formula.
(21)$$\left[\begin{array}{c} \beta_{x}(k)\\ \beta_{y}(k)\\ \beta_{z}(k) \end{array}\right]=\left[\begin{array}{ccc} -1/\tau_{x} & 0 & 0\\ 0 & -1/\tau_{y} & 0\\ 0 & 0 & -1/\tau_{x} \end{array}\right]\left[\begin{array}{c} \beta_{x}(k-1)\\ \beta_{y}(k-1)\\ \beta_{z}(k-1) \end{array}\right]+\left[\begin{array}{c} n_{\beta x}\\ n_{\beta y}\\ n_{\beta z} \end{array}\right]$$

The state equation of the system can be derived from the gyroscope random drift formula and the discrete quaternion state variable equation.
(22)$$x(k)=\left[\begin{array}{cc} F(k-1) & 0_{4\times 3}\\ 0_{3\times 4} & \tau (k-1) \end{array}\right]x(k-1)+\left[\begin{array}{c} 0_{4\times 1}\\ n_{\beta } \end{array}\right]$$

The system utilizes accelerometers and magnetometers to compensate for errors caused by gyro drift because the angular velocity measured by the gyroscope is used for the quaternion update. Therefore, the quaternion is calculated using the output of the accelerometer, and the magnetometer is selected as the measurement variable. If the measured values of the design accelerometer and magnetometer are, respectively, $f={\left[f_{x},f_{y},f_{z}\right]}^{T}$ and $h={\left[h_{x},h_{y},h_{z}\right]}^{T}$, then the attitude angle calculated using the measured value is shown in the following formula.
(23)$$\{\begin{array}{l} \phi =\arctan(\frac{f_{y}}{f_{z}})\;\\ \theta =-\arctan(\frac{f_{x}}{\sqrt{{\left(f_{y}\right)}^{2}+{\left(f_{y}\right)}^{2}}})\\ \psi =\arctan(\frac{h_{z}\sin\phi -h_{y}\cos\phi }{h_{x}\cos\theta +h_{y}\sin\phi \sin\theta +h_{z}\cos\phi \sin\theta }) \end{array}$$

From the attitude angle Formula (23), the measured value is calculated according to the quaternion formula.
(24)$$\left[\begin{array}{c} \begin{array}{l} q_{0} \end{array}\\ \begin{array}{l} q_{1} \end{array}\\ \begin{array}{l} q_{2} \end{array}\\ q_{3} \end{array}\right]=\left[\begin{array}{c} \cos\frac{\phi }{2}\cos\frac{\theta }{2}\cos\frac{\psi }{2}+\sin\frac{\phi }{2}\sin\frac{\theta }{2}\sin\frac{\psi }{2}\\ \sin\frac{\phi }{2}\cos\frac{\theta }{2}\cos\frac{\psi }{2}-\cos\frac{\phi }{2}\sin\frac{\theta }{2}\sin\frac{\psi }{2}\\ \cos\frac{\phi }{2}\sin\frac{\theta }{2}\cos\frac{\psi }{2}-\sin\frac{\phi }{2}\cos\frac{\theta }{2}\sin\frac{\psi }{2}\\ \cos\frac{\phi }{2}\cos\frac{\theta }{2}\sin\frac{\psi }{2}-\sin\frac{\phi }{2}\sin\frac{\theta }{2}\cos\frac{\psi }{2} \end{array}\right]$$

Considering noise is unavoidable in the actual environment, therefore, the observation equation of the system is as follows.
(25)z(k)=H(k)x(k)+v(k)
where v(k) is the observation noise of the system, H(k) is the measurement noise, the expression is as follows.
(26)$$H_{k}=[I_{4\times 4},0_{4\times 3}]$$

The state equation and the observation equation of the system constitute the filtering model of the system. The small coaxial unmanned helicopter is suitable for working in unknown environments. Electromagnetic radiation, hard magnetic material, soft magnetic material and other magnetic fields interfere. Magnetometers are susceptible to interference, resulting in large noise in the measured values. System noise parameters greatly affect the performance and stability of the unscented Kalman filter.

### 2.4. Filtering Model

The statistical characteristics of the observed noise in the system’s observation equation in the framework of the UKF filter algorithm are unchanged. When the noise changes, there is some error in the model, which leads to an increase of the filter estimation error. Therefore, the accuracy of the azimuth measurement cannot be guaranteed for the coaxial UAV under interference. In order to improve the adaptive ability of the UKF algorithm to noise interference, an adaptive estimation algorithm is combined with UKF to adjust the covariance of the measured noise according to the real-time measurements of the two side variables and improve the filtering accuracy [12,13,14]. The EKF algorithm structure diagram as shown in Figure 2.

Initializing the state, assuming that the initial value filter is as shown in Equation (27).
(27)$$\{\begin{array}{l} {\hat{x}}_{0}^{+}=E(x_{0})\\ P_{0}^{+}=E[(x_{0}-{\hat{x}}_{0}^{+}){\left(x_{0}-{\hat{x}}_{0}^{+}\right)}^{T}] \end{array}$$

The EKF filtering step, time prediction update estimation Formula (28) is shown.
(28)$${\hat{x}}_{k|k-1}=f({\hat{x}}_{k-1|k-1},u_{k})$$
(29)$$P_{k|k-1}=F_{k}P_{k-1|k-1}F_{k}^{T}+Q_{k}$$

In Equation (29) above, $Q_{k}$ is the process noise covariance matrix at time *k*. In the measurement update formula, $P_{k|k-1}$ is the corresponding covariance matrix at time *k*.
(30)$${\tilde{y}}_{k}=z_{k}-h\left({\hat{x}}_{k|k-1}\right)$$
(31)$$S_{k}=H_{k}P_{k|k-1}H_{k}^{T}+R_{k}$$
where $H_{k}\;$is the measurement Jacobian matrix at time *k*, and $R_{k}$ is the observation noise covariance matrix at time *k* in Equation (31).
(32)$$K_{k}=P_{k|k-1}H_{k}^{T}S_{k}^{-1}$$ where the $K_{k}$ Kalman gain is calculated and ${\hat{x}}_{k|k}$ is updated by the observed variable.
(33)$${\hat{x}}_{k|k}={\hat{x}}_{k|k-1}+K_{k}{\tilde{y}}_{k}$$

The error covariance is updated, and the measurement update equation uses the values of the observed variables to correct the state and covariance estimates. The Accord ratio matrix of the state transition matrix and the measurement matrix is shown in Equation (34).
(34)$$F_{k}=\frac{\partial f}{\partial x}|{\hat{x}}_{k-1|k-1,u_{k}}$$
where $H_{k}=\frac{\partial h}{\partial x}{|}_{{\hat{x}}_{k|k-1}}$ is the process Accord ratio matrix.

## 3. Adaptive UKF Filtering Algorithm

### 3.1. Adaptive Estimation of Measurement Noise

The measured value derived from the real value is often used to estimate the optimal value of the noise parameter of the system. The adaptive filtering method based on the maximum likelihood criterion can estimate the changes of the statistical characteristics of the system noise in real time through the covariance matrix of the system state and the covariance matrix of the measurement noise to ensure that the filter can adapt to the changes better. The maximum likelihood criterion is estimated from the perspective of the maximum probability of measurement occurrence, which takes into account both the change of new interest rate and the change of covariance of new interest rate [15,16]. According to the maximum likelihood criterion, the adaptive measurement noise variance matrix is satisfied in Formula (35).
(35)$${\hat{R}}_{k}=\frac{1}{N}\sum_{i=k-N+1}^{k}\varepsilon_{i}\varepsilon_{i}{}^{T}-H_{k}P_{k|k-1}H_{k}^{T}$$
where *N* is the number of Windows and represents the amount of new information used. In nonlinear systems, it is very difficult to calculate ${\hat{R}}_{k}$ using the adaptive measurement equation formula, thus it is necessary to simplify and improve it.

When *N* = 1, the measured value of the measurement noise covariance matrix is Formula (36).
(36)$$R_{k}^{+}=\varepsilon_{k}\varepsilon_{k}{}^{T}-H_{k}P_{k|k-1}H_{k}^{T}$$

The weighted average of $R_{k}^{+}$ and the predicted value $R_{k}^{-}$ of the measurement noise covariance matrix at the current time is taken as the estimated value of the measurement noise covariance matrix at the current time, i.e.,
(37)$${\hat{R}}_{k}=R_{k}^{+}-\mu (R_{k}^{-}-R_{k}^{+})$$
where μ is an adjustable parameter, and the new information sequence contributes more to the measurement noise covariance estimate. By substituting ${\hat{R}}_{k}$ in the above formula into the filtering algorithm, the measurement noise covariance matrix can be estimated in real time, and then the filtering gain can be adjusted, which is the adaptive ability of the filtering algorithm to noise changes. The parameter μ is determined by the simulation method.

### 3.2. Filtering Algorithm

Based on the filtering model, the principle of sensor data fusion using the adaptive UKF filtering algorithm is shown in Figure 3.

According to the established model and the implementation process of the UKF algorithm, the specific steps of sensor data fusion using adaptive UKF filtering are as follows.

(1) Determine the statistical characteristics of the initial filtering state [17].

It is necessary to determine the initial quaternion, the initial value of the gyroscope drift and the error covariance matrix of the initial state vector. The calculation formula is as follows.
(38)$${\hat{x}}_{0}=E(x_{0})$$
(39)$$P_{0}=E[(x_{0}-{\hat{x}}_{0}){\left(x_{0}-{\hat{x}}_{0}\right)}^{T}]$$

(2) Select three particle sigma point sampling strategies to calculate sigma points and their weights.

The sampling strategy of proportion correction was adopted, and sigma points were selected according to the following formula.
(40)$$\{\begin{array}{l} \xi_{0}^{k}={\hat{x}}_{k}\\ \xi_{i}^{k}={\hat{x}}_{k}+{\left(\sqrt{(n+\chi )P_{k}}\right)}_{i},\\ \xi_{i+n}^{k}={\hat{x}}_{k}-{\left(\sqrt{(n+\chi )P_{k}}\right)}_{i}, \end{array}i=1,2,\ldots ,n$$

The weights of the corresponding mean expectation and variance are A and B, as follows.
(41)$$\{\begin{array}{l} W_{0}^{m}=\frac{\chi }{n+\chi }\\ W_{0}^{c}=W_{0}^{m}+1+\beta -\alpha^{2},\\ W_{i}^{c}=\frac{1}{2(n+\chi )} \end{array}i=1,2,\ldots ,2n$$
where *n* is the state dimension, $\chi =\alpha^{2}\left(n+\kappa \right)-n$, where κ is usually 0, *α* usually takes a very small positive number, and *β* has the best value of 2 in Gaussian distribution. UT transforms the approximation of the true expectation and variance.

(3) Time update.

According to the choice of sigma points and weights, to determine the state variable step predictive value ${\hat{x}}_{k|k-1}\;$and step prediction error covariance matrix $P_{k|k-1}$:(42)$$\gamma_{i}^{k|k-1}=f(\xi_{i}^{k-1})$$
(43)$${\hat{x}}_{k|k-1}^{}=\sum_{i=0}^{2n}W_{i}^{m}\gamma_{i}^{k|k-1}$$
(44)$$P_{k|k-1}=\sum_{i=0}^{2n}W_{i}^{c}(\gamma_{i}^{k|k-1}-{\hat{x}}_{k|k-1}){\left(\gamma_{i}^{k|k-1}-{\hat{x}}_{k|k-1}\right)}^{T}+Q_{k-1}$$
where $Q_{k-1}$ is the system state noise covariance matrix.

(4) Measurement update. Calculate the measurement variable step forecast $Z_{k|k-1}$ and variance $P_{zk}$, as well as the mutual covariance matrix $P_{x_{k}z_{k}}$.
(45)$$z_{k|k-1}=H_{k-1}x_{k|k-1}$$
(46)$$P_{zk}=H_{k-1}P_{k|k-1}H_{k-1}{}^{T}+R_{k}$$
(47)$$P_{x_{k}z_{k}}=P_{k|k-1}H_{k-1}^{T}$$ where $R_{k}$ is the measurement noise covariance matrix, which is the key parameter calculated in the adaptive UKF filtering. It can be calculated by using the estimated value of the measurement noise covariance matrix.

After obtaining the new quantity measurement $Z_{k}$, obtain the optimal estimate of the system state ${\hat{x}}_{k}$ and the state covariance matrix $P_{k}$.

The Kalman filter gain is as follows.
(48)$$K_{k}=P_{x_{k}z_{k}}P_{z_{k}}{}^{-1}$$

The new rate sequence is as follows.
(49)$$\varepsilon_{k}=z_{k}-z_{k|k-1}$$

The optimal estimate of the system state matrix is as follows.
(50)$${\hat{x}}_{k}={\hat{x}}_{k|k-1}+K_{k}\varepsilon_{k}$$

The state covariance matrix is as follows.
(51)$$P_{k}=P_{k|k-1}-K_{k}P_{z_{k}}K_{k}{}^{T}$$

## 4. Attitude Control Test Methods

In order to complete the attitude control experiment task, a coaxial UAV experimental prototype is built by using machining and 3d printing technology. The coaxial UAV experimental prototype is composed of upper and lower rotors, fuselage, rotor structure bearing, landing gear and so on. The whole body of the prototype adopts a barrel structure shape made of carbon fiber pipe, and the power of the prototype adopts the structure of the common axis counter-rotor. The two steering engines and the aluminum alloy central shaft controlling the tilting disc maintain a plane, and the plane of the tilting disc is perpendicular to the lift force of the lower propeller of the UAV; thus, the human-machine attitude can be changed by changing the tilting angle of the tilting disc. The power part of the UAV consists of two motors and two steering gear [18,19]. The up and down propeller rotation is realized by the motor speed regulating and driving the gear transmission. The attitude control of the UAV is accomplished by two steering engines and the central axis controlling the tilt plate plane. The electronic part of the UAV is mainly composed of flight control, electrical adjustment, steering gear, receiver, data transmission, GPS and other parts. As the core control unit of the UAV, the flight control completes data fusion, attitude calculation, attitude control and position control. The receiver is responsible for the communication between the operator and the UAV, and the PPM modulation is responsible for the receiving and sending of RC remote command at the ground end [20]. Relevant parameters of the UAV and UAV test platform are shown in Table 1.

In order to verify the effectiveness of the attitude control method, a complete test platform for a coaxial twin-rotor flight test prototype was built. The platform includes experimental prototype, external sensor, external processor, current regulator power supply, ground station communication system and other parameter analysis software modules [21]. As shown in the Figure 4, the flight simulation test platform of the co-axial folded twin-rotor aircraft adopts the cantilever suspension design as a whole, which saves the freedom of rotation in one direction and connects the top of the UAV with a safety rope. The universal joint of the common axis UAV base and the support rod structure is fixed, and the whole UAV and the universal joint connection part can rotate and move up and down freely. The physical measurement platform includes a pneumatic lift and power test, torque and speed test, tilt plate control angle test and vibration test. The test project can complete the test methods of lift, vibration acceleration and attitude data of the co-axial dual-rotor aircraft, and it can detect the attitude data of the UAV through the IMU of the internal flight control and the external sensor [22]. The vertical flight process of the co-axial twin-rotor aircraft is mainly affected by the aerodynamic force generated by the rotor rotation and its own gravity. The lift force of the UAV can be detected by collecting the base tension sensor. According to the analysis of the different motion states of the UAV, the change of throttle volume is adjusted to verify the influence of the vibration characteristics on IMU [23].

The experiment is divided into static and dynamic testing. Static testing mainly tests the relationship between the UAV attitude vibration acceleration and the gas volume, compared with the no trace of the Kalman filtering algorithm, the Kalman filter algorithm and the gyroscope integral algorithm to solve the attitude angle. The effect of dynamic testing mainly verifies the no trace of the accuracy of the Kalman filter to solve attitude angle [24].

### 4.1. Test Scheme

The prototype uses an inertial measurement unit (IMU) consisting of a three-axis accelerometer, a three-axis gyroscope (MPU6500), a three-axis reluctance meter (HMC5583L) and a high-performance STM32 microprocessor. The raw data measured by the sensor are sampled with the sampling frequency of 100 Hz by the microprocessor, and the raw data are sent to the industrial control computer through the serial port without filtering. The extended Kalman filter designed by MATLAB is used to process the data on the computer in order to obtain accurate attitude information. Finally, the experimental results are shown by curve description, and the calculation results are compared and analyzed. In order to verify the effectiveness of the proposed algorithm, five groups of experiments are carried out. The measurement value of the sensor of the test system in the static state is collected through the static measurement platform, as shown in the following Figure 5. The attitude angle information is measured and collected through the inertial sensor and sent to the main controller module through the serial bus [25,26].

### 4.2. Vibration Test

According to the experimental requirements, the duty cycle of the input quantity is adjusted by external remote control, and the RC signal is collected by oscilloscope to ensure the accurate quantization input of the signal. Considering the input range {0.2σ,0.4σ,0.6σ,0.8σ,0.9σ}, the industrial control computer completes the quantization sampling and processing of the attitude information of the IMU and external sensor.

The test results {0.2σ,0.4σ,0.6σ,0.8σ,0.9σ} show the attitude vibration curve as shown in Figure 6. When the throttle input quantity gradually increases, the throttle control quantity ranges from 0.2σ to 0.9σ. The vibration amplitude of the UAV becomes larger and larger, and the amplitude in the XYZ direction increases with the throttle volume. In Figure 6: for the coaxial unmanned XYZ, three directions of vibration amplitude, and the relationship between the input figure, the test process of uniform stability of input sampling 1500 sampling points, and remote control throttle sigma from sigma duty ratio 0.2 to 0.9 between the platform under different excitation vibration amplitude of the data, it can be seen that the vibration of the UAV vibration amplitude increases with the increase of the excitation source. ag5 is the response value of 0.9σ excitation, which shows that the amplitude reaches a maximum range of 1.1 g. The influence on IMU can be reduced by analyzing the characteristics of the measurement noise and establishing a noise model.

According to the processed image of the vibration acceleration data measured in the test as shown in Figure 7, it can be seen that the amplitude range of vibration acceleration under the control of the roll (pitch, yaw) parameters and overall parameters is mostly within ±1 g. Among them, there is a maximum acceleration peak of roll motion vibration, which is not more than 1.25 g. According to the acceleration of vibration and the frequency of acquisition, the amplitude of vibration obtained can be indirectly calculated according to Equation (11) below. The frequency of the data collected by the vibration sensor is 100 Hz, the estimated amplitude of the vibration is about ±0.15 mm and the amplitude of the roll motion is not more than 0.2 mm. According to the test results, it can be seen that the controller has good controllability, indicating that it can control the flight of the aircraft [27,28,29].

### 4.3. The Static Test

The UAV is still placed on a horizontal platform. With a roll angle and pitching angle of 0, the coaxial UAV attitude Angle measurement environment is a relatively closed environment. There is no wind. Using the extended Kalman filtering and no trace of Kalman filtering algorithms, the UAV static and accelerometer were not influenced by the body vibration and the acceleration of gravity disturbance. The attitude calculated by accelerometer and the magnetometer is taken as the measurement value, the angular velocity information measured by the gyroscope is fused and the static attitude is solved with high accuracy. The roll angle calculated by the fusion algorithm is more accurate. The variation curve of vibration acceleration in pitching motion is shown in Figure 8. The vibration acceleration curve of the rolling motion is shown in Figure 9. The variation curve of yaw vibration acceleration is shown in Figure 10.

When the throttle value is 0.5σ, the change of attitude angle is detected, and the comparison results of the static roll angle are calculated by the EKF fusion algorithm. The roll angle calculated by the EKF filter has angle drift, and the error is close to 1, while the error of the UKF filter algorithm is less than 0.5. When the throttle value is 0.5σ, the change of the attitude angle is detected, and the comparison results of the static pitch angles are calculated by the EKF fusion algorithm. The roll angle calculated by the EKF filter has angle drift, and the error is close to 2, while the error of the UKF filter algorithm is less than 0.6. When the throttle value is 0.5σ, the change of the attitude angle is detected, and the comparison results of the static yaw angle are calculated by the EKF fusion algorithm [30,31]. The roll angle calculated by the EKF filter has angle drift, and the error is close to 0.5, while the error of the UKF filter algorithm is less than 0.4.

### 4.4. Dynamic Flight Experiment

In order to complete the prototype flight test and verify the attitude following effect, the flight test was carried out in the outdoor environment of the campus of the University of Science and Technology, where the GPS signal was unobstructed and there was no wind disturbance. The coaxial UAV flew 90 m in the X direction and 27 m in the Y direction, and the flight altitude was controlled at about 2 m. After the flight was completed, the flight log of 400 s was selected and analyzed. From the flight log, the actual flight trajectory and expected value approximation of the UAV could be obtained. For good follow-up attitude, the maximum flying speed is 1.32 m/s and the maximum flying altitude is 2.1 m. Figure 11, Figure 12 and Figure 13 shows the expected value and actual value of dynamic flight XYZ direction. Figure 14 shows the expected value and actual value of dynamic flight path. Figure 15 shows the actual flight diagram.

## 5. Conclusions

Aiming at the problem of error accumulation and large interference of a single attitude measurement device for coaxial UAV attitude measurement, a data fusion method of EKF and UKF is proposed to ensure a more accurate attitude angle. The dynamic performance and attitude angle test platform of the coaxial UAV half-in-the-loop flight was established. The platform analyzes the performance of UAVs at different rotor speeds and attitude angles. The main noise sources and main random errors of the coaxial UAV are mainly analyzed. Aiming at the attitude calculation problem of the coaxial dual-rotor UAV, an optimized adaptive unscented Kalman filter algorithm and an extended Kalman filter algorithm are proposed to advance the model [32,33,34]. In contrast, the algorithm uses the gradient descent algorithm to reproduce and adjust the process noise covariance to optimize the state prediction and estimation, which effectively reduces the error of the state solution. In the process of attitude calculation, the attitude quantity is represented by quaternion, and the variation trend of roll angle, pitch angle and yaw angle is analyzed in the outdoor flight attitude verification test. The flight results show that the attitude algorithm has good attitude estimation characteristics, indicating that the research strategy of this subject has sufficient engineering application value.

## Figures and Tables

**Figure 1 sensors-22-09572-f001:**
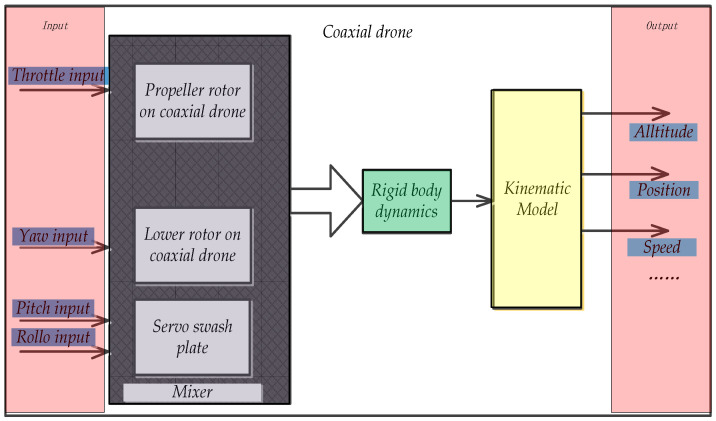
Structural model diagram of coaxial twin-rotor UAV.

**Figure 2 sensors-22-09572-f002:**
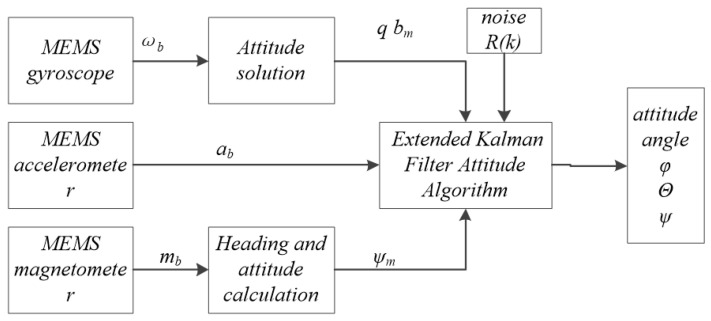
EKF algorithm structure diagram.

**Figure 3 sensors-22-09572-f003:**
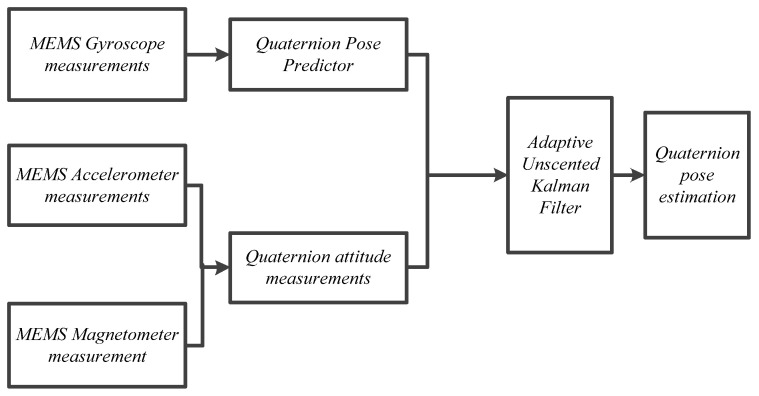
The adaptive UKF algorithm.

**Figure 4 sensors-22-09572-f004:**
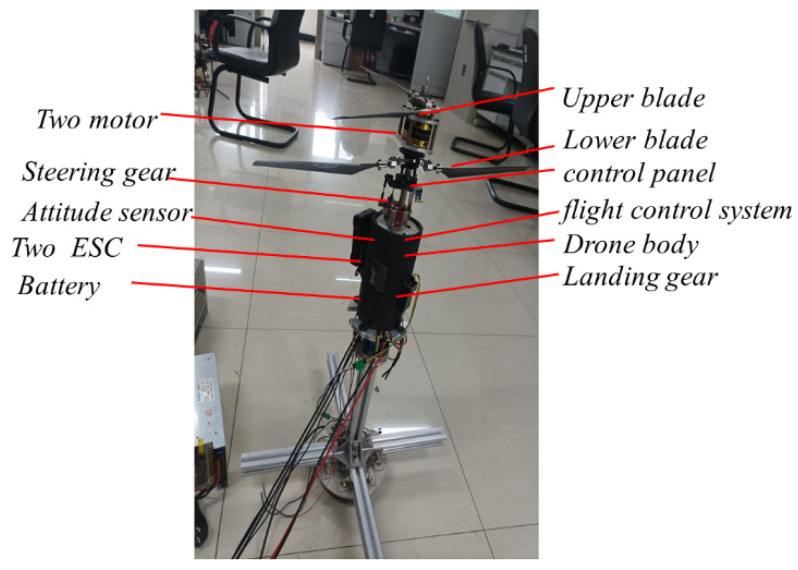
Structure diagram of the prototype of a coaxial twin-rotor.

**Figure 5 sensors-22-09572-f005:**
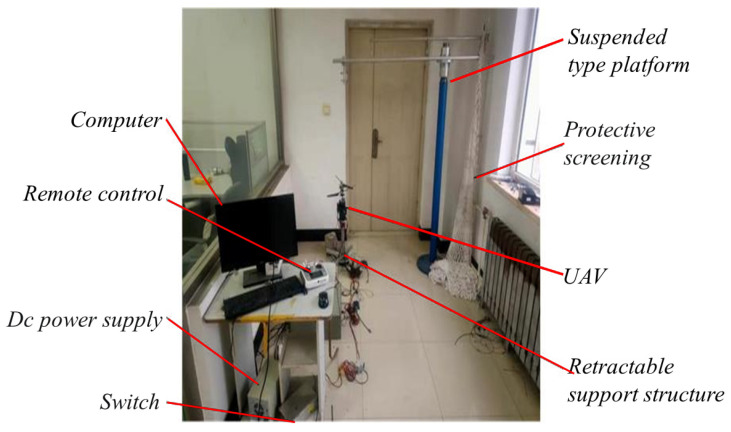
Static test platform diagram of the coaxial twin-rotor.

**Figure 6 sensors-22-09572-f006:**
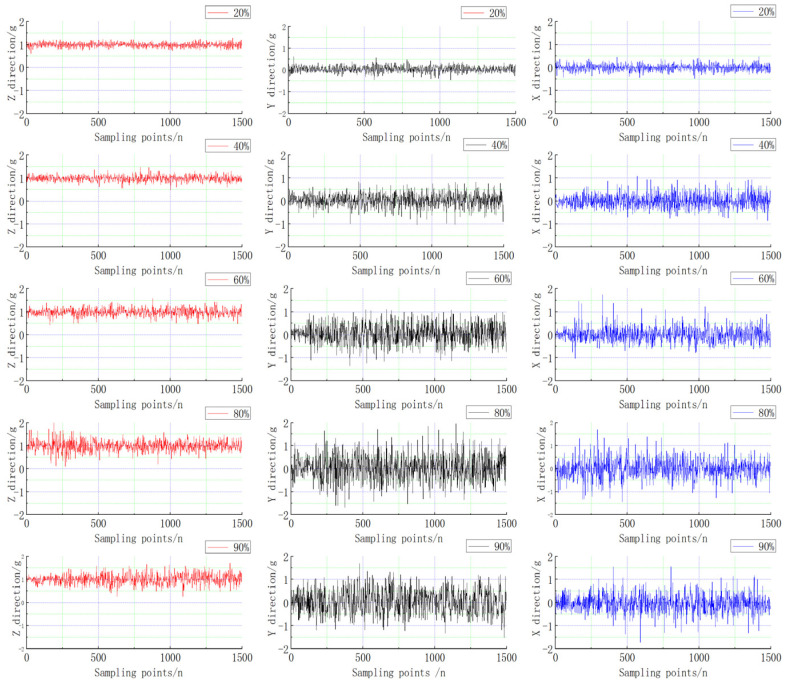
Variation of vibration acceleration in the XYZ direction.

**Figure 7 sensors-22-09572-f007:**
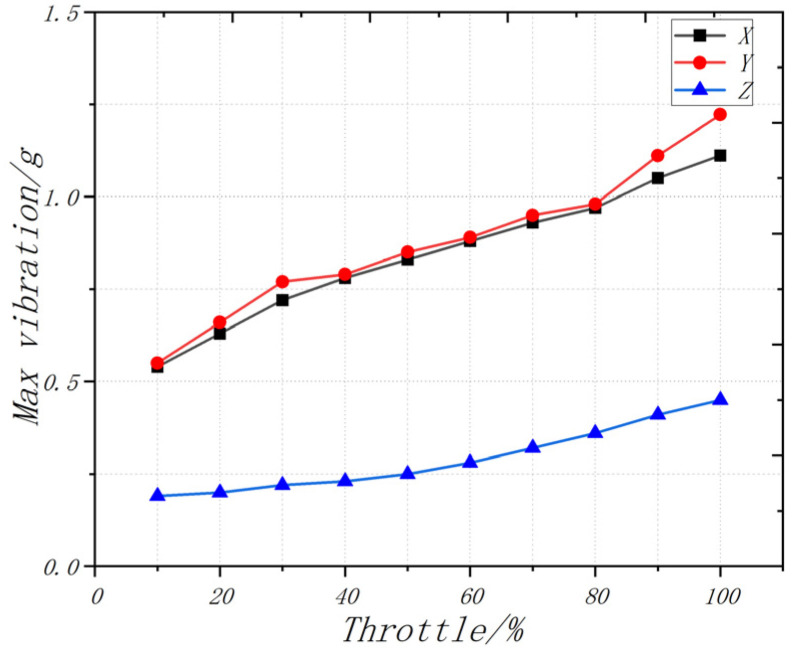
Relationship between throttle volume and maximum vibration peak value.

**Figure 8 sensors-22-09572-f008:**
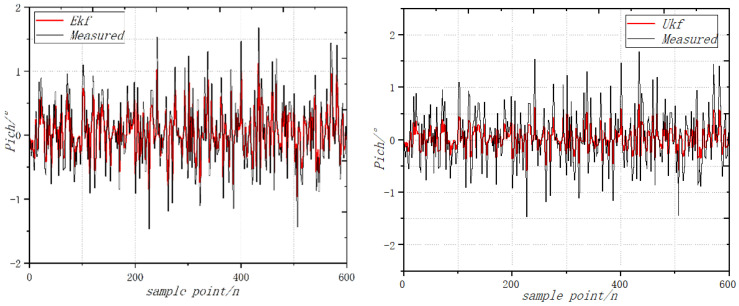
Variation curve of vibration acceleration in pitching motion.

**Figure 9 sensors-22-09572-f009:**
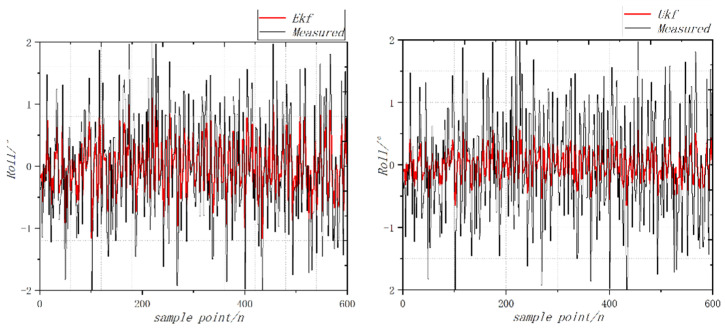
Curve of vibration acceleration of rolling motion.

**Figure 10 sensors-22-09572-f010:**
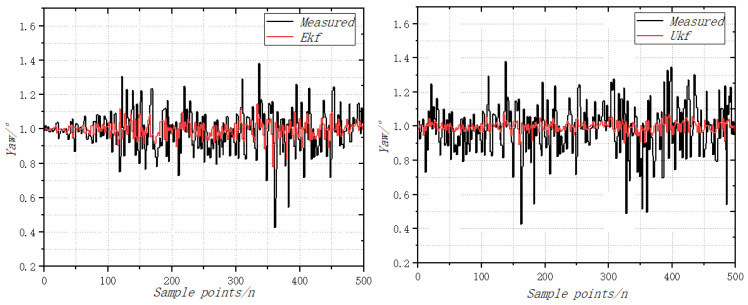
Variation curve of yaw vibration acceleration.

**Figure 11 sensors-22-09572-f011:**
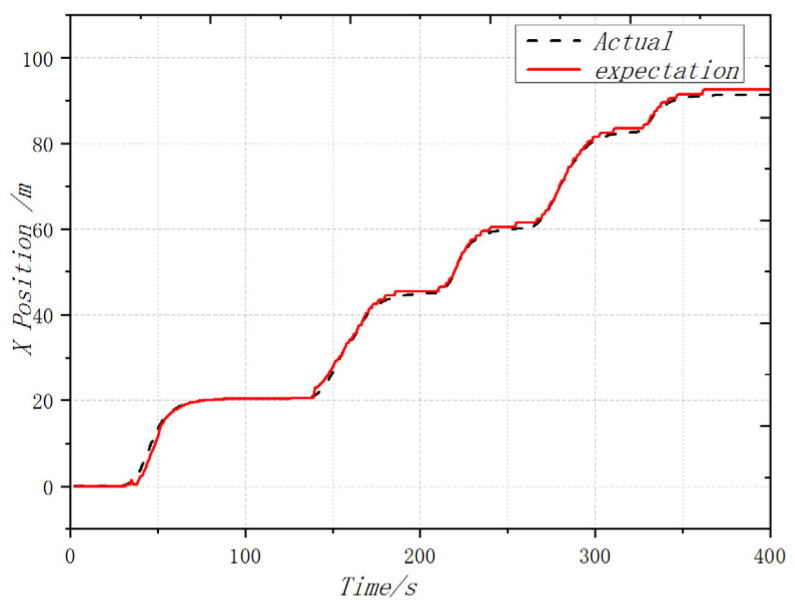
Expected value and actual value of dynamic flight X direction.

**Figure 12 sensors-22-09572-f012:**
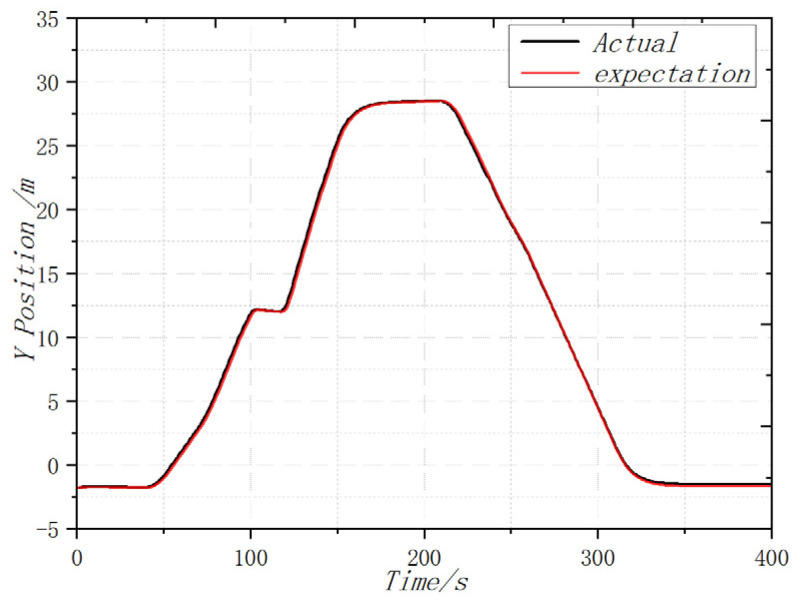
Expected value and actual value of dynamic flight Y direction.

**Figure 13 sensors-22-09572-f013:**
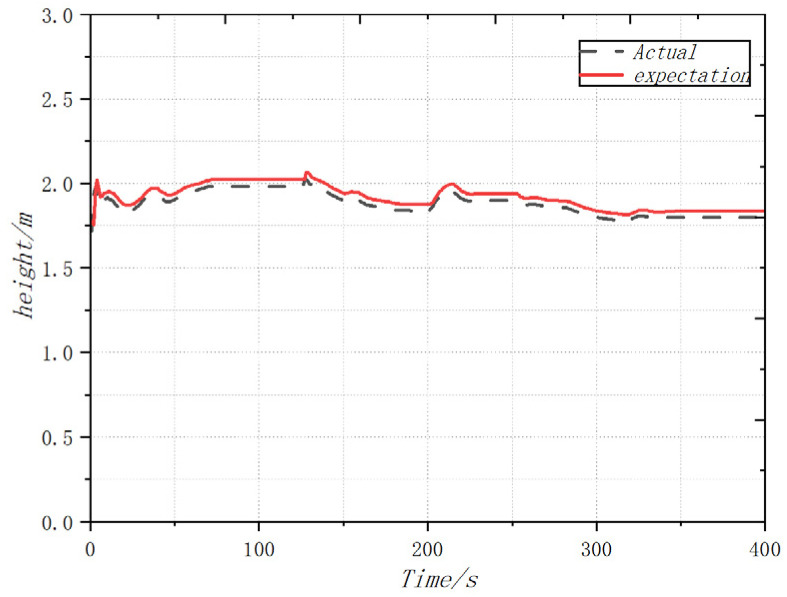
Expected value and actual value of dynamic flight Z direction.

**Figure 14 sensors-22-09572-f014:**
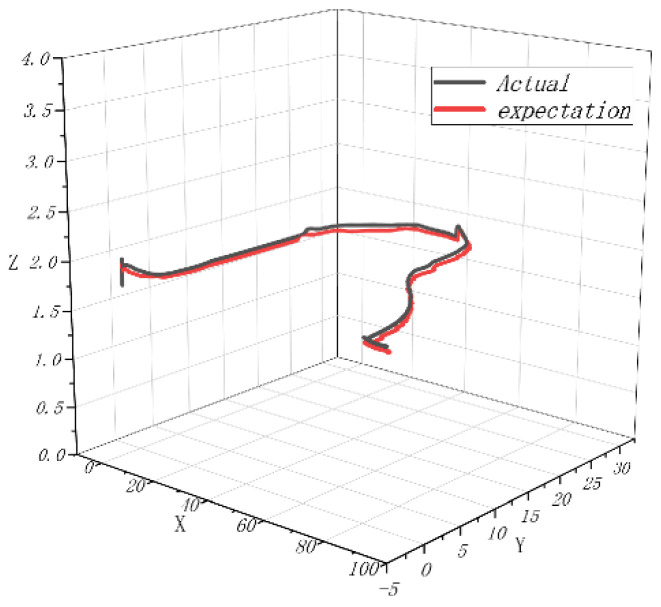
Expected value and actual value of dynamic flight path.

**Figure 15 sensors-22-09572-f015:**
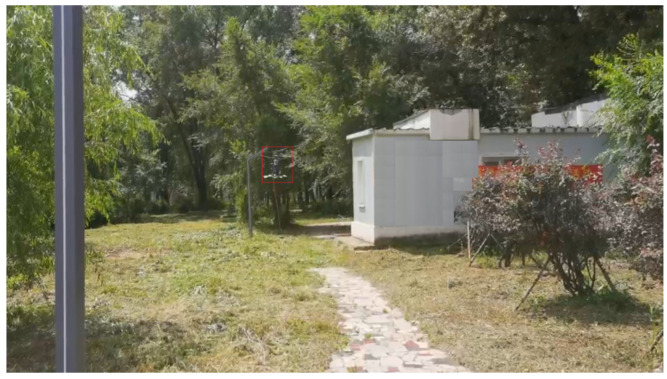
Actual flight test figure.

**Table 1 sensors-22-09572-t001:** Table of relevant parameters of the UAV and UAV test platform.

UAV Test Platform	Platform Related Parameters
UAV	Coaxial twin rotor with diameter of 80
UAV remote control	Fs-i6s
Battery	1800 mah 35 c 4 s14.8 v
ESC	Platinum-25A-6S-V4
Paddle	19.5 cm ∗ 5 cm ∗ 0.5 cm
Electric motor	HKII-2213-14 3200kv
Steering engine	KST DS215MG V3.0
Attitude sensor	Six axis wireless Bluetooth attitude sensor
Vibration sensor	Digital MEMS vibration sensor
Computer	CPU: Intel Core I7 8700, Memory: 8 GB (1333 MHz), Hard disk: solid state drive 240 G
Pressure sensor	DAYSENSOR DYLF-102
Power	15 V DC power supply

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
