# Peer review of "Research on Attitude Detection and Flight Experiment of Coaxial Twin-Rotor UAV"

_sensors, 2022, doi:10.3390/s22249572_

Round 1

Reviewer 1 Report

Have no further comments.

Author Response

Dear Editor, First of all, we would like to thank the editor and the reviewers for the time and effort they have spent on reading and commenting on our work. We are grateful to be granted the opportunity to revise the manuscript and we believe that by following the reviewers suggestions we have improved the manuscript. We hope we have addressed all issues to the satisfaction of the reviewers and the editor.

Reviewer 2 Report

In this paper, the research on attitude detection and flight experiment of coaxial twin-rotor UAV is developed. This paper can be accepted after minor revision.

(1) The motivation and the contribution of this paper should be given point by point in introduction;

(2) The flowchart of the proposed algorithm should be given more clearly;

(3) The parameters of the experimental devices should be added in a Table;

(4) EKF and UKF are employed in this paper. However, these two algorithms are normal. Author should explain why the two algorithms are used here, the comparison with some other novel algorithms should be analyzed, such as "A Hybrid IMM Based INS/DVL Integration Solution for Underwater Vehicles", "Dual-optimization for a MEMS-INS/GPS system during GPS outages based on the cubature Kalman filter and neural networks";

(5) A formal conclusion should be added;

(6) Quantitative comparison results should be given for the comparison figures;

(7) What's the algorithm complexity of the proposed algorithm? Is it can be used in real application?

Author Response

Dear reviewers, Thank you for your kind patience. We have revised our manuscript carefully by following the guidance provided by the editor and reviewers. Here are the point-by-point responses for the reviewers’ comments.

(1)We have started to add the purpose and main research contribution of this article one by one in the introduction section.  And at the end of this section, the main description of the paper is added.

The paper analyzes the kinematics and dynamics characteristics for the coaxial attitude estimation method.  Establish a hardware-in-the-loop simulation attitude angle test platform to analyze the mechanical vibration and attitude angle performance of UAVs.  Experimental analysis of the mechanical vibration characteristics of different rotor speeds. The EKF and UKF filtering algorithm models are established, and the trend of attitude changes over time is verified in static and dynamic conditions, respectively.  The UKF attitude estimation method is better than the EKF attitude estimation method, and the test flight experiment verifies that the proposed algorithm has better attitude estimation results for the coaxial UAV.

In order to solve the problem that the single sensor of the coaxial UAV cannot accurately measure the attitude information, an attitude estimation algorithm based on unscented Kalman filter information fusion is proposed.

(2)We have made the algorithm flow diagram in the paper more clear.

(3)We have made a table for the parameters of the experimental device, and made a more detailed description of the parameters.

(4)We have analyzed the direct differences between related algorithms, and we show that this algorithm has better pose estimation.

(5)We would like to thank you for giving us the opportunity to revise our manuscript. According to suggestions, we corrected the text and figures. We believe that this revised version will satisfy the readers of your journal.

(6)We again thank a lot to the Reviewers and Editor for correcting some unclear wording and sentences in our manuscript to improve the readability and language quality. We really appreciate their kind help.

(7)Thank you for your interest in our study and we appreciate your detailed and professional recommendations. We paid close attention to the precious comments and clarified some vague points or expressions in the main text. UKF can be regarded as a Kalman filter based on UT technology.  The Kalman filter algorithm uses UT transformation to deal with the non-linear transmission of mean and covariance to the one-step forecasting equation.  The UKF approximates the probability density distribution of the nonlinear function, and uses a series of posterior probability densities to determine the approximate state of the sample, without the need for the derivation of the Jacobian matrix. In practical applications, for nonlinear systems, UKF has higher accuracy and stability.

Reviewer 3 Report

The content of the article is consistent with the scientific area of the journal MDPI Sensors. The subject raised by the authors is current and so far rarely noticed by other authors publishing in this area.
The issue described may in the future contribute to improving the efficiency of the automation of the unmanned aerial or control system.
The paper has an original, scientific character, related to the research on attitude detection and Flight Experiment. In this paper, the conventional location estimation methods are analyzed and the extended Kalman filter algorithm and the uncentered Kalman filter algorithm are introduced. This result can provide a reference for optimizing the control parameters and flight control strategies of a coaxial folding twin-rotor aircraft.
For a better clarification, please edit your paper as follows: 1. Extend the text of the manuscript (e.g. introduction or conclusion) with specific results in the world and Europe, - Improve the quality of the paper by presenting the results of publications by researchers and experts who are involved in this field and are registered in world databases (wos). These are e.g: Path planning optimization of six-degree-of-freedom robotic manipulators using evolutionary algorithms or Investigation of Snake Robot Locomotion Possibilities in a Pipe, thanks. 2. figure 1 should be contrasting and readable,
3. conclusions and future work should be extended to contain practical applications based on research described in this paper - expand references,
4. highlight the course of dependencies/relations in figure No.  7 - the green color is indistinct ,
5. Modify the mathematical expression (formula) No: 6 and 35.
I recommend publishing the post after the proposed modifications.

Author Response

(1)Dear Editor and Reviewers, Thanks for your comments concerning our paper. Those comments are all valuable and very helpful for revising and improving our paper, as well as the important guiding significance to our research. We have studied comments carefully and have made modification which we hope meet with approval. Revised portion are marked in red in the paper.

(2)We have studied comments carefully and have made revisions to meet with approval. We have modified and improved the picture 1. Revisions are marked in red in the paper.

(3)The paper analyzes the kinematics and dynamics characteristics for the coaxial attitude estimation method.  Establish a hardware-in-the-loop simulation attitude angle test platform to analyze the mechanical vibration and attitude angle performance of UAVs.  Experimental analysis of the mechanical vibration characteristics of different rotor speeds.  The EKF and UKF filtering algorithm models are established, and the trend of attitude changes over time is verified in static and dynamic conditions, respectively.  The UKF attitude estimation method is better than the EKF attitude estimation method, and the test flight experiment verifies that the proposed algorithm has better attitude estimation results for the coaxial UAV.

(4)Dear reviewers, Thank you for your kind patience. We have revised our manuscript carefully by following the guidance provided by the editor and reviewers. Here are the point-by-point responses for the reviewers’ comments.
(5)Once again, thank you for your work and valuable comments on this manuscript. We have made relevant modifications and processing.

Reviewer 4 Report

The paper needs some revisions in order to improve its scientific quality and merits.

1. The Abstract section should be improved by adding 1-2 sentences at the end of it, pointing out the achieved results in terms of numbers. Moreover, it is not very well and clear stated the aims of this paper in the Abstract.

2. The Introduction section. At the end of this section, the paper structure description will be useful if added.

Before this sub-section part, it is also recommended to add 2-3 phrases declaring the aim and research contribution of this paper.

3. Section 2.1 and 2.2 are well-known in literature and therefore, are useless in this paper. I suggest that the Authors to add practical implementation of "attitude" measurement sensors/systems used for a coaxial twin rotor UAV based platform.

4. Figure 4. Use red color for descriptive lines. The same for Figure 5.

5. Add at the end of the paper a Section for Conclusions. Describe in this Section the main conclusions of this work.

Author Response

Dear, Reviewer ,Thank you for your kind review of our paper. I can agree with you. I have revised the manuscript according to your advices in the following reply. Please review our paper once again. The added part of the thesis is as follows in blue.

(1)First, I added a clear purpose of the research in the first part of the abstract of the article. Second, I made a relevant comparison at the end of the abstract and added the main experimental results.

Aiming at the problem that a single sensor of coaxial UAV cannot accurately measure attitude information. In this paper, a pose estimation algorithm based on unscented Kalman filter information fusion is proposed.

The analysis results show that the throttle amount is between 0.2σ and 0.9σ. The UKF pose estimation is better than that of EKF, with higher accuracy and less than 0.6° filter algorithm error. The experimental analysis provides a reference for the optimization of control parameters and flight control strategies of coaxial folding twin-rotor aircraft.

(2)The Introduction section. At the end of this section, the added sections are shown in blue.

The paper analyzes the kinematics and dynamics characteristics for the coaxial attitude estimation method.  Establish a hardware-in-the-loop simulation attitude angle test platform to analyze the mechanical vibration and attitude angle performance of UAVs.  Experimental analysis of the mechanical vibration characteristics of different rotor speeds. The EKF and UKF filtering algorithm models are established, and the trend of attitude changes over time is verified in static and dynamic conditions, respectively.  The UKF attitude estimation method is better than the EKF attitude estimation method, and the test flight experiment verifies that the proposed algorithm has better attitude estimation results for the coaxial UAV.

The measurement accuracy of the attitude information of the coaxial UAV directly affects the navigation accuracy of the speed and position of the aircraft. Gyroscopes, accelerometers, and magnetometers are often used as attitude measurement devices. Gyroscopes have cumulative errors, while accelerometers and magnetometers are subject to greater external interference. Effective data fusion is required to ensure a more accurate attitude horn.

(3)The input of the coaxial dynamic model is the tension and torque provided by the propeller, and the output is the speed and angular velocity of the coaxial dual rotors. The input of the kinematics model is the output of the dynamics model, that is, the speed and angular velocity of the UAV, and the output is the position and attitude of the UAV.

(4)Thank you for your professional comments and suggestions, which are very helpful for us to improve the manuscript.  We have carefully considered the reviewers; critical comments and thoughtful suggestions and have made figure(4)(5) changes to the manuscript accordingly.  The revised sections of our manuscript are marked and highlighted.

(5)The added part of the thesis is as follows.

Aiming at the problem of error accumulation and large interference of a single attitude measurement device for coaxial UAV attitude measurement, a data fusion method of EKF and UKF is proposed to ensure a more accurate attitude angle.  The dynamic performance and attitude angle test platform of the coaxial UAV half-in-the-loop flight was established.  The platform analyzes the performance of UAVs at different rotor speeds and attitude angles.  The main noise sources and main random errors of the coaxial UAV are mainly analyzed.  Aiming at the attitude calculation problem of the coaxial dual-rotor UAV, an optimized adaptive unscented Kalman filter algorithm and an extended Kalman filter algorithm are proposed to advance the model.  In contrast, the algorithm uses the gradient descent algorithm to reproduce and adjust the process noise covariance to optimize the state prediction and estimation, which effectively reduces the error of the state solution.  In the process of attitude calculation, the attitude quantity is represented by quaternion, and the variation trend of roll angle, pitch angle and yaw angle is analyzed in the outdoor flight attitude verification test.  The flight results show that the attitude algorithm has good attitude estimation characteristics, indicating that the research strategy of this subject has sufficient engineering application value.

Round 2

Reviewer 4 Report

no comments.

Author Response

Dear Reviewer, First of all, we would like to thank the editor and the reviewers for the time and effort they have spent on reading and commenting on our work. We are grateful to be granted the opportunity to revise the manuscript and we believe that by following the reviewers suggestions we have improved the manuscript. We hope we have addressed all issues to the satisfaction of the reviewers and the editor.